# Use of patient-centred outcome measures alongside the personal wheelchair budget process in NHS England: A mixed methods approach to exploring the staff and service user experience of using the WATCh and WATCh-Ad

**Lorna Tuersley**[1], **Naa Amua Quaye**[1], **Kalpa Pisavadia** [1], **Rhiannon Tudor Edwards**[1]\*, **Nathan Bray** [2]

1 Centre for Health Economics and Medicines Evaluation (CHEME), School of Health Sciences, Bangor University, Gwynedd, United Kingdom, 2 Academy for Health Equity, Prevention and Wellbeing (AHEPW) School of Health Sciences, Bangor University, Gwynedd, United Kingdom

\* Kalpa.pisavadia@bangor.ac.uk

## Abstract

### Background and objective

Personal wheelchair budgets (PWBs) are offered to everyone in England eligible for a wheelchair provided through the National Health Service (NHS) to support their choice of equipment. The WATCh (Wheelchair outcomes Assessment Tool for Children) and related WATCh-Ad for adults are patient-centred outcome measures (PCOMs) developed to help individual users express their main outcome needs when obtaining a wheelchair and rate their satisfaction with subsequent outcomes after receiving their equipment. Use was explored in a real-world setting, aiming to produce guidance for use alongside the PWB process.

### Methods

Three wheelchair service provider organisations across four sites participated. Staff and users completed surveys about their experience of assessments using the WATCh and/or WATCh-Ad. Selected patients were interviewed after receipt of their equipment, and staff were interviewed after experiencing a number of assessments. Thematic analysis was undertaken using the tool, survey and interview data. Results of pre- and post-equipment provision were presented graphically.

### Results

Information on 75 assessments by 15 staff was obtained. Three-quarters of users or their carers rated the use of the tools in the assessment process as 'helpful' or 'very helpful'. Staff reported that the WATCh or WATCh-Ad had been considered 'useful' in developing

**Data Availability Statement:** All relevant data are within the paper and its Supporting Information files.

**Funding:** Funding for the lead author (LT) was received from NHS England who had commissioned the initial development of the PCOM evaluated in this study. Further funding for the researchers contributing towards this study (LT and KP) Was received from Health and Care Economics Cymru, which is supported by Health and Care Research Wales funding, via Welsh Government. Staff from NHS England formed part of the study team, overviewing the design of the research and seeking expressions of interest from participating CCGs. They had no role in the data collection and analysis, the decision to publish, or the preparation of the manuscript.

**Competing interests:** I have read the journal's policy and the authors of this manuscript have the following competing interests: the funding for the the lead author for the study was received from NHS England who had commissioned the initial development of the PCOM study evaluated in the study. This does not alter our adherence to PLOS ONE policies on sharing data and materials.

**Abbreviations:** CCG, Clinical Commissioning Group; COVID-19, COrona VIrus Disease 2019; GDPR, General Data Protection Regulations; IoM, Institute of Medicine; LREC, Local Research Ethics Committee; NHS, National Health Service; NIHR, National Institute for Health Research; n/s, Not stated; NWAG, National Wheelchair Advisory Group; OT, Occupational Therapist; PCOM, Patient-Centred Outcome Measure; PT, Physiotherapist; PWB, Personal Wheelchair Budget; R&D, Research and development; SU, Service user (term used in responses); UK, United Kingdom; WATCh, Wheelchair outcomes Assessment Tool for Children; WATCh-Ad, WATCh tool for Adults; WSQ, Wheelchair Satisfaction Questionnaire.

individual care plans in around 1 in 3 cases and affected the prescription in 1 in 4 cases. Concerns were expressed about the length of time taken to administer the tools in clinic. However, some staff noted this reduced with more hands-on experience and by providing the tools to users in advance of the appointment.

## Conclusions

The WATCh and WATCh-Ad PCOMs are suitable for routine use by wheelchair service providers to assist the assessment process. It is recommended that tool materials are provided in advance to users/carers and that staff are allowed time to develop their ways of working with them.

## Introduction

In 2021/22, 24% (over 16 million people) of the United Kingdom (UK) population reported having some form of disability. Mobility impairment is the second most common cause of disability in the UK, with 43% of people with a disability reporting some form of mobility impairment, equating to almost 7 million people [1]. In England, more than 660,000 people were registered with wheelchair services in June 2023, including nearly 68,000 children. The average Clinical Commissioning Group (CCG) received over 1,000 new and re-referrals for wheelchairs in January to March 2023 [2]. Research has highlighted that ineffective provision of wheelchairs increases costs to users in terms of potential harm to health of delayed or ill-fitting equipment, increases costs to the NHS in terms of wasted resources and increases costs to society as a whole if users are not fully integrated into society [3].

The NHS has responded to demands to increase the efficiency of service provision and address the needs of individual users to provide wheelchairs that meet individual needs in a number of ways. These include collection of quarterly data on service provision by CCGs [4] and the introduction of personal wheelchair budgets (PWB) to help provide everyone with access to a wheelchair that meets their individual health and well-being needs and goals [5]. The legal right to a personal health budget of people who access wheelchair services, whose posture and mobility needs impact their wider health and social care needs came into force in December 2019 [6].

The WATCh (Wheelchair outcomes Assessment Tool for Children) tool was developed as part of an NHS England-funded programme of research to develop Patient-Centred Outcome Measures (PCOMs) for use with children and young people [7]. At the time of developing the WATCh, none of the outcome measures available for use by therapists and assistive technology providers in the UK were specifically aimed at children and young adults requiring a wheelchair. The authors worked with children, young people, and their carers to understand the wide range of outcomes from using their wheelchair that were important to them. The resulting tool aimed to help wheelchair users and clinical staff identify key outcomes prospectively and measure changes in outcome satisfaction after receiving new wheelchair equipment. The WATCh lists 16 predetermined outcome areas, from which users can select the five outcomes of most importance to themselves at their assessment and rank them in order of importance (Part A). There is also the option to describe an 'Other' outcome should they feel there is something important that is not covered in the predetermined list. Users then state, in their own words, what they wish to achieve for each outcome. In Part B, they rate their current satisfaction with these outcomes, using a 5-point scale of 'smiley faces', ranging from 1 ('Very

unsatisfied') to 5 ('Very satisfied'). Some months after receipt of the equipment, a follow-up tool, Part C, is sent to the user to complete, requesting a re-rating of their satisfaction against their chosen outcomes. The two scores can be compared by the clinician. Any unchanged or more negative updated scores can then be the focus for discussion and subsequent adjusted or different provision.

Ongoing work by the authors to develop the MoBQoL-7D preference-based measurement of mobility-related quality of life has shown that the general outcomes desired by adults are closely related [8], hence the wording of this tool was adapted for use with adults as WATCH-Ad. The tools can be provided in a paper-based format (see S1 File) or electronically (see https://cheme.bangor.ac.uk/watch.php).

The WATCh and WATCh-Ad go some way to address the assertion made by Kenny and Gowran (2014) that no single outcome measure for wheelchair and seating provision addressed an intervention's contribution to the activity and participation of the individual and captured the influence of the entire service provision on the quality of life of the individual [9].

Most measures in use assess patients using fixed predetermined areas rather than focusing on the user's own needs and preferences, for example, the Therapy Outcome Measures (TOM) [10] assesses patients against predefined areas using pre-coded levels of achievement. Other measures aimed at users of wheelchairs use predetermined areas with a focus on functionality and skills, such as the Wheelchair Users Functional Assessment (WUFA) [11], the Functioning Everyday with a Wheelchair (FEW) [12, 13], the Activities Score for Kids (ASK) [14], and The Functional Mobility Assessment (FMA) [15], also adapted for use with children as the FMA-FC [16].

The Quebec User Evaluation of Satisfaction with Assistive Technology (QUEST) [17] evaluates levels of satisfaction with aspects of the service or the technology, and while it is relevant to wheelchair users, it only captures satisfaction with what has already been provided. Similarly, the Wheelchair Satisfaction Questionnaire (WSQ) aims to provide data from wheelchair users to manufacturers and providers about their existing chair. It asks for satisfaction ratings with predefined aspects of their current wheelchair, and does not probe which aspects are of most importance to them [18]

The Psychosocial Impact of Assistive Devices (PIADS) [19, 20] focuses on self-rated functional independence, well-being and quality of life of the patient. Initially developed for adults, it lists 26 predetermined areas for rating, while the children's adaptation utilises a five-point 'smiley faces' Likert-type scale of agreement with 15 short statements to assess the constructs of Competence, Adaptability, and Self-esteem [20].

Other tools use interviews to determine outcomes defined by patients or in collaboration with patients. However, they are not aimed at wheelchair users specifically, for example, the interview-based Goal Attainment Scaling (GAS) measure [21, 22] and the Canadian Occupational Performance Measure (COPM) [23], both of which have been used in paediatric research [24].

The Wheelchair Outcome Measure (WhOM) [25] and the subsequent adaptation for use with young people as the WhOM-YP [26] have an aim and approach very similar to that of the WATCh and WATCh-Ad and also incorporate clinical aspects.

The authors believe that use of the WATCh tools allows the user to express in their own words the most important aspects of their lives that they wish their new mobility equipment would be able to help them with, in a way not captured by the other tools noted above. The scoring data is unique to the user, and results are not intended to be used for comparisons of interventions, in the way that a tool such as the MoBQoL would be used.

## Aims

The uptake of any new treatment or process relies on successful implementation. NHS England has sought to embed PCOMs into the PWB pathway and funded the research described to help develop guidelines for roll-out nationwide. Our aim was to assess outcome achievement at the level of the patient, referred to here as the service user, by comparison of satisfaction ratings with their key outcome areas before and after provision of equipment, and identify aspects of use of the tools important to them and the service provider in order to maximise wider implementation. A further aim was to investigate the practicalities of use and the resource implications of introducing a PCOM into the PWB process. This is the first study to assess the use of a PCOM across several providers alongside the introduction of PWBs.

## Methods

A mixed methods approach was used. Separate staff and service user survey questionnaires aimed to obtain quantitative and limited qualitative information on each assessment, to allow comparison of the acceptability of the process and perceived advantages and disadvantages to both staff and service users. In addition, telephone interviews were to be carried out towards the end of the study among a range of service users following receipt of their new wheelchair and with service staff who had experience working with the tools. Data from the use of WATCh or WATCh-Ad was reviewed to assess the responses from users and to compare an individual's satisfaction scores before and after receipt of their prescribed equipment.

The researchers worked with the NHS England personal health budgets team and the National Wheelchair Advisory Group (NWAG), which includes wheelchair service providers and users, to develop the scope of the work and inform the questionnaires and interview schedules. A study management team involving lead contacts from the participating organisations met throughout the study to oversee progress. The study involved human subject research approved by the local Health and Care Research Wales Research and Ethics Committee Wales REC 5 (20/WA/0007). Informed consent was to be obtained in writing from the participants, or where relevant their parent/carer or proxy. In November 2019, the NHS England personal health budgets team invited expressions of interest from wheelchair services in England to participate. Selection was based on providing a mix of experience of use of the WATCh and or WATCh-Ad and the implementation of PWBs, geographic location and whether they were NHS-staffed or managed through independent contractors. The four sites from three provider organisations had differing levels of experience: one with prior experience of both the PWB process and of the tools, provided through an independent contractor in the North East of England (Site A), two smaller sites in the Midlands run by a second independent contractor organisation, with prior experience of the PWB process but not of the tools (Sites B and C), and one provided directly through the NHS in the South East with no experience of either (Site D). Table 1 shows the information provided in response to the expressions of interest from the selected organisations.

Participation packs for staff and potential participants were provided to the sites in advance, sufficient for each anticipated assessment. Staff packs contained information on the study, consent forms to obtain written informed consent and a post-assessment survey. Service user packs included information sheets tailored to specific age groups for children (as requested by the ethics committee who had reviewed the WATCh development research previously) [7] and for adults (aged over 16), forms to obtain written informed consent user or proxy consent as appropriate, the relevant WATCh or WATCh-Ad PCOM form, and a post-assessment survey. Packs were to be given to the service user and the assessing staff member just prior to the assessment (see S2 File).

**Table 1. Participating organisations and service sites–data provided in expressions of interest.**

| | | Site A | Site B* | Site C* | Site D | Total number of service users/referrals across all sites |
|---|---|---|---|---|---|---|
| Location of service | | North of England | Midlands | Midlands | South East | |
| Type of contractor | | Independent | Independent | Independent | NHS | |
| Start use of PWB | | February 2019 | April 2019 | May 2019 | December 2019 | |
| Start use of WATCh | | February 2019 | - | - | - | |
| Adults/ quarter | New | 425 | 206 | 119 | 530 | 1280 |
| | Re-referrals | 545 | 283 | 132 | 356 | 1316 |
| Children/ quarter | New | 33 | 21 | 18 | 90 | 162 |
| | Re-referrals | 110 | 52 | 64 | 79 | 305 |
| Total number of adult referrals | | 970 | 489 | 251 | 886 | 2596 |
| Total number of child referrals | | 143 | 73 | 82 | 169 | 467 |
| Total new referrals | | 458 | 227 | 137 | 620 | 1442 |
| Total re-referrals | | 655 | 335 | 196 | 435 | 1621 |
| Total number of referrals | | **1113** | **562** | **333** | **1055** | **3063** |

*Note.* *Sites B and C were combined as they were managed by same organisation and had similar overall users to A and D

Staff recorded the time taken by the service user or carer to complete the tool and who completed it. They also noted any difficulties experienced and whether the staff member was able to resolve them. They were invited to comment on the usefulness of the tool, any impact it had on the prescription, and the PWB option selected.

The user survey questionnaires asked service users for more personal details, their reason for needing a wheelchair, and the time since their previous assessment (if any). They also asked about the practicalities of completing the tool, including time to complete, how easy it was to understand, any questions they had, and whether they felt that any outcome areas were missing. Written informed consent was sought for information about their assessment and any follow-up data being made available to the research team in an anonymised format. Contact details were only required if they consented to a follow-up interview. Where the assessor site considered a user to be unable to complete responses (e.g. due to age, capacity or health issues), their parent/guardian or carer was invited to submit responses and complete consultee forms as appropriate. Staff and user surveys were handed into the clinic co-ordinator in sealed envelopes for return to the researcher to protect the anonymity of the responses.

The WATCh and WATCh-Ad PCOM data was shared with the researchers to allow them to quantify the outcomes selected and their scores and any changes between the assessment and follow-up outcomes. The qualitative data on reasons for choice of outcomes was also reviewed. As not all users involved in the study were expected to gain sufficient experience with their equipment to be able to complete the follow-up WATCh PCOM within the time constraints of the study, staff in sites with prior use of the tool were asked about their experience of follow-up during their interviews. Sites were asked to provide anonymised, non-identifiable data from prior assessment and follow-up scores for descriptive statistical analysis.

A number of the staff and service users who had provided written informed consent to be interviewed over the telephone were contacted towards the end of the study. The interview guide contained open-ended questions exploring their survey responses and the use of the tools in the PWB process. Staff interviewees included a manager and therapists from each

organisation. Service users were sampled purposively from those consenting to interviews, to encompass those having their first assessment and those who had previously been given a wheelchair, adults, children and whether the tool was completed by the user (with assistance if needed) or their parent/or carer.

### Impact of the COVID-19 pandemic upon the methods

On 19th March 2020, the Director of Community Health for NHS England and NHS Improvement wrote to community service providers, directing them as to how capacity could be released to support the COVID-19 pandemic preparedness and response [27]. This led to a significant reduction in wheelchair clinic provision. Due to staff redeployment, Site D was unable to start recruitment before this date. In addition, non-COVID-19 related research involving human participants was paused during this period. Permission was given by funders, Bangor University's School ethics approval and the Local Research Ethics Committee (L-REC) to extend the study until 31st March 2021. At the end of May 2020, the National Institute for Health Research (NIHR) issued guidance on the Restart Framework [28], and in July 2020, activities were restarted at the sites contracted to independent organisations, who had previously recruited participants. Site A continued to triage patients prior to attending clinic, but it was agreed that Sites B and C would cease recruitment as sufficient numbers had been identified elsewhere. Staff at Site D were able to start planning for recruitment following a review with their organisations' research and development departments. Updated information for adults relating to General Data Protection Regulation requirements (GDPR) were approved by the authors' institution and ethics committee in December 2020 to allow interviewing to start.

Data collection ended on 16th February 2021 to allow for review of data and follow-up of any queries within the extension period. At Site D, although 24 assessments were planned during this period, use of the tools was only possible in 19: five were deemed unsuitable by the clinician as they were home visits for vulnerable, shielding patients where face-to-face contact time had to be kept to a minimum.

The protocol had stated that each organisation should aim for a total of 100 assessments to be carried out, assuming a consent to interview rate of 10%, based on responses to a questionnaire mailing in the original WATCh study [7]. Although the number of participants was much lower than this due to the pandemic, the consent rate for interviews was 72%, and represented the majority of the participant types required.

Despite the very high consent rate for interviews given in the patient surveys, of the first six consenting users contacted, only two interviews could be arranged. Two service users did not respond to phone calls, one declined to take part, and one interview was cancelled due to family circumstances due to the pandemic. The two interviews that did take place gained limited additional information compared with the users' surveys. Given this low uptake and that the delay since the original assessment in most cases would require recall of quite distant events, it was agreed that no further attempts would be made to contact users. Consent from staff was high, and six interviews with staff took place as planned, which were able to provide additional information to their surveys based on information about use in practice and their reflections on this (See S3 and S4 Files).

### Analysis

Data from the service users' surveys and the associated staff survey was entered into Microsoft Excel for each service user. The datasets were explored and analysed to provide descriptive statistics including by subgroups of site, user versus staff reports, adults versus children, and first-time assessments versus re-referred patients. Unless otherwise stated, percentages are based on

75 participants, the total of assessments completed by staff. In five cases (7%), the users themselves did not want their survey data included, so they have been included as 'not stated' (n/s) responses.

Qualitative data obtained through the surveys and the interviews was transcribed and analysed for emerging themes assisted by use of a Microsoft Excel spreadsheet. Results are presented by the type of information obtained in order to compare between staff and service users.

# Results

## Participants recruited

Information on a total of 90 potential uses of the tools in service user assessments was returned by the sites: 27 from Site A, 39 from Sites B and C combined, and 24 from Site D. Ten planned assessments at Site C were not included as five users or their carers did not want to take part; two felt the survey and tool were too long, two appointments were cancelled, and one failed to provide any written consent to the study. Five potential uses in home visits at Site D after the re-start of the study did not take place due to the user being considered vulnerable and was shielding.

Fifteen different staff members across the services provided a total of 75 surveys on assessments using the tool. Seventy surveys were completed by users or their carers, and 67 consented to the information from their PCOMs tools (WATCh, $n = 19$; WATCh-Ad, $n = 48$). Demographic data from the service users and staff members is presented in Table 2.

## Staff

Staff returning surveys on the assessments included ten occupational therapists (OT), two OT/ Clinical leads, one physiotherapist (PT), and one research engineer (Table 3). One did not state their occupation. In addition, Site A employed a PWB liaison officer who went through the Tool questions before the assessment. Interviews were carried out with six members of staff between December 2020 and February 2021, including the PWB liaison officer at Site A.

**Service users.** Staff surveys reported use of the tools with 53 adults and 22 children aged under 16, with ages ranging from 2 to 90 years. Table 2 shows the breakdown by age and gender for each site.

The 70 service users' own surveys included 49 adults and 21 children. Eight adults and twelve children were first-time attendees to wheelchair services, eleven adults and nine children had been previously seen by a wheelchair service within the past 12 months, twelve adults and four children had been seen by a wheelchair service between 1 and 5 years ago, and nine adults had been seen longer than five years ago.

Service users' reasons for requiring wheelchair services were varied, and many only provided general statements relating to difficulty getting around. Table 3 shows the range of underlying reasons split by adults and children and by whether this was their first assessment or not:

Despite the smaller number of participants than originally planned, there was a high rate of consent to interview, given by 53 users or their carers (70%), although information to make contact was missing in three cases. These represented most of the respondent types required across all sites by age, site, whether new or returning user or the tool was completed by user or carer. The exception was children under 16 completing the tools themselves.

## Survey findings

Staff surveys asked about use of the tools in practice, including questions about any additional time taken compared to an assessment without using them. They were also asked about any

**Table 2. Participating staff and service users.**

| User group | Site A Adults | Children | Site B/C Adults | Children | Site D Adults | Children | Total number of adults | Total number of children | Total |
|---|---|---|---|---|---|---|---|---|---|
| **Potential uses of WATCh PCOMs** | **25** | **2** | **27** | **12** | **12** | **12** | **64** | **26** | **90** |
| **Staff surveys** | **25** | **2** | **19** | **10** | **9** | **10** | **53** | **22** | **75** |
| n OT (n surveys) | 6 (20) | 1 (2) | 5 (16) | 3 (6) | 1 (1) | - | 12 (37) | 4 (8) | 12 (45) |
| n PT (n surveys) | | | | | 1 (8) | 1 (10) | 1 (8) | 1 (10) | 1 (18) |
| n Manager/ Clinical Lead (n surveys) | 1 (5) | | | 1 (1) | | | 1 (5) | 1 (1) | 2 (6) |
| n Other (n surveys) | | | 2 (3) | 1 (3) | | | 2 (3) | 1 (3) | 2 (6) |
| Total staff surveyed | 7 (25) | 1 (2) | 7 (29) | 5 (10) | 2 (9) | 1 (10) | 16 (53) | 7 (22) | 17 (75) |
| Mean age male user years (SD) | 45.6 (17.7) | 7 (0) | 58.9 (15.5) | 7.6 (5.0) | 57.0 (17.58) | 9.5 (4.4) | 51.5 (16.52) | 8.0 (4.24) | 37.4 (24.96) |
| Mean age female user years (SD) | 51.1 (22.1) | 6 (0) | 48.8 (29.9) | 8.0 (5.4) | 51.8 (17.91) | 7.5 (3.8) | 51.5 (23.43) | 7.6 (4.07) | 38.5 (27.74) |
| Mean age gender n/s years (SD) | - | - | 76.3 (7.2) | 5 (0) | 42.5 (20.5) | 10.5 (2.1) | 64.2 (19.77) | 10.5 (2.12) | 50.8 (29.95) |
| PCOM completed by User | 13 | 0 | 6 | 1 | 3 | **1** | 22 | 2 | **24** |
| Completed by Carer | 3 | 2 | 6 | 10 | **2** | **7** | **11** | 19 | 30 |
| Completed by Staff | 3 | - | 3 | - | - | - | **6** | - | 6 |
| Completed by other including with assistance | 8 | - | 4 | - | **1** | - | **13** | - | 13 |
| Blank | 1 | - | - | - | - | 1 | 1 | 1 | 2 |
| **Staff interviews** | | | | | | | | | |
| n OT (n surveys) | 1 (5) | | 1(9) | 2 (3) | | | 3 (15) | 1 (2) | 3 (17) |
| n PT (n surveys) | | | | | 1 (8) | 1 (10) | 1 (8) | 1 (10) | 1 (18) |
| n Manager/ Clinical Lead (n surveys)[1] | 1 (5) | | | 1 (1) | | | 1 (5) | 1 (1) | 1 (6) |
| n Other | 1[2] | 1[2] | | | | | | | |
| Total staff Interviews | 2 | 1 | 2 | 2 | 1 | 1 | 6 | 4 | 6 |
| **User surveys** | **25** | **2** | **16** | **10** | **8** | **9** | **49** | **21** | **70** |
| Male | 11 | 1 | 10 | 5 | 3 | 4 | 24 | 10 | 34 |
| Female | 14 | 1 | 6 | 4 | 4 | 4 | 24 | 9 | 33 |
| Gender n/s* | - | - | - | 1 | 1 | 1 | 1 | 2 | 3 |
| Age range male | 21–72 | 7 | 27–77 | 2–13 | 44–77 | 5–15 | 21–77 | 2–15 | 2–77 |
| Age range female | 19–86 | 6 | 22–90 | 2–15 | 18–71 | 5–13 | 18–90 | 2–15 | 2–90 |
| Age range gender n/s | - | - | 68–81 | 6 | 28–57 | 9–12 | 28–81 | 9–12 | 9–81 |
| PCOM completed by Self | 10 | - | 3 | - | 5 | 1 | 18 | 1 | 19 |
| PCOM completed by Carer | 4 | 2 | 5 | 10 | 3 | 7 | 12 | 19 | 31 |
| PCOM completed by Staff | - | - | 4 | - | - | - | 4 | - | 4 |
| Other including user or carer-assisted | 10 | - | 4 | - | - | - | 14 | - | 14 |
| No information on who completed | 1 | - | - | - | - | 1 | 1 | 1 | 2 |
| Written informed consent to interview | 19 | 2 | 13 | 8 | 5 | 4 | 37 | 14 | 51 |
| New Service Users | 4 | 2 | 7 | 1 | - | 1 | 11 | 4 | 15 |
| Previous Service Users | 15 | - | 6 | 7 | 5 | 3 | 26 | 10 | 36 |
| New–Self consent | 4 | - | 6 | - | - | - | 10 | - | 10 |
| New–Carer consent | - | 2 | 1 | 1 | - | 1 | 1 | 4 | 5 |
| Previous–Self consent | 13 | - | 2 | - | 3 | - | 18 | - | 18 |
| Previous–Carer Consent | 2 | - | 4 | 7 | 2 | 3 | 8 | 10 | 18 |

(*Continued*)

**Table 2.** (Continued)

| User group | Site A | | Site B/C | | Site D | | Total number of adults | Total number of children | Total |
|---|---|---|---|---|---|---|---|---|---|
| | Adults | Children | Adults | Children | Adults | Children | | | |
| **Tool data** | **24** | **2** | **17** | **9** | **7** | **8** | **48** | **19** | **67** |
| Male | 10 | 1 | 10 | 5 | 3 | 3 | 23 | 9 | 32 |
| Female | 14 | 1 | 6 | 3 | 3 | 4 | 23 | 8 | 31 |
| Gender n/s[3] | - | - | 1 | 1 | 1 | 1 | 1 | 2 | 3 |
| Age range male | 21–73 | 7 | 27–74 | 2–13 | 44–76 | 5–15 | 21–77 | 2–15 | 2–77 |
| Age range female | 19–86 | 6 | 22–90 | 2–8 | 49–71 | 5–13 | 19–90 | 2–13 | 2–90 |
| Age range gender n/s | - | - | - | 5 | n/s | 9 | 5–9 | n/s | 5–9 |

[1]Clinical Lead = OT

[2]Respondent was 'Broker'–recorded tool information but did not perform clinical assessment

[3]Includes two who did not complete user surveys but consented to share tool information

problems with service users' ability to complete the tool and their own ability to deal with any issues arising. These were compared with the user survey responses to similar questions. Figures showing the findings in more detail are presented in S5 File.

The mean time taken to complete the tool by users was reported to be 12.2 minutes by both staff (range 5 minutes to 45 minutes, n = 58 reporting this) and users themselves (range 2

**Table 3. Underlying need for wheelchair reported by new or returning user.**

| | First-time wheelchair service user | | Returning wheelchair service user | | First or returning n/s | |
|---|---|---|---|---|---|---|
| | Reason | *n* | Reason | *n* | Reason | *n* |
| **ADULT** | General | 4 | General | 10 | Cerebral Palsy | 1 |
| | Amputee | 1 | Multiple Sclerosis | 3 | Frederick ataxia | 1 |
| | Arthritis | 1 | Cerebral Palsy | 3 | Multiple Sclerosis | 1 |
| | Falls | 1 | Amputation | 2 | | |
| | Fibromyalgia | 1 | Pain | 2 | | |
| | Parkinson's | 1 | Arthritis | 1 | | |
| | Spinal condition | 1 | Arthrogryposis/Multiple sclerosis | 1 | | |
| | | | Broken hips | 1 | | |
| | | | Ehlers Danlos Syndrome | 1 | | |
| | | | Huntington's | 1 | | |
| | | | Myotonic Dystrophy | 1 | | |
| | | | Nerve damage | 1 | | |
| | | | Paraplegia | 1 | | |
| | | | Spinal arthritis | 1 | | |
| | | | Stroke | 1 | | |
| | | | Stroke/Arthritis | 1 | | |
| | **TOTAL** | **12** | **TOTAL** | **32** | **TOTAL** | **9** |
| **CHILD** | Cerebral Palsy | 1 | General | 8 | | |
| | General | 1 | Cerebral Palsy | 3 | | |
| | Hypermobility | 1 | Autism/ global development | 1 | | |
| | Pain | 1 | Complex | 1 | | |
| | Seizures | 1 | Dandy-Walker Syndrome | 1 | | |
| | | | Hypermobility | 1 | | |
| | **TOTAL** | **6** | **TOTAL** | **15** | **TOTAL** | **0** |

minutes to 45 minutes, $n$ = 55). In terms of the person completing the tool forms, staff reported this was either the user or their carer in 54 assessments (72%) while a slightly smaller proportion 67% (n = 50) of user/carer surveys stated it to be the user or carer without help from staff.

Staff were asked about any problems with completing the tools and whether they were able to deal with them, as Yes/No options. In 35 (47%) cases, no problems were reported. Comments made by staff relating to nineteen (25%) instances of giving 'No' or n/s responses related to the inability to complete the tool without explanation ($n$ = 5; 7%), administration of the form, for example, omitting to bring to clinic or client felt rushed ($n$ = 5; 7%), difficulty in choosing outcomes or setting goals ($n$ = 4; 5%) and communication issues such as dyslexia, writing difficulties and need for a family member to translate ($n$ = 3; 4%). In one case (1%), the staff member felt it was irrelevant as the service user only needed adjustment to the headrest. In another (1%), the parent of a child with a terminal condition found the tool to be insensitive.

In contrast, the majority of users/carers ($n$ = 61; 81%) felt that it was easy to understand, and 75% ($n$ = 56) reported no problems with using the tools. Comments from seven users/carers who felt it was not easy to understand included that it was too long or not relevant for what they were being assessed for and one user/carer commented the tool was not easy to understand because it was completed over the telephone. Five users/carers responded 'yes' to the question about whether anything was missing; where made, comments related to the general approach of the tool or to the user's complex clinical condition, rather than their expectations from a wheelchair.

Staff surveys also asked whether use of the tools was useful or not in that assessment. Where rated, they were stated to be useful in 35% ($n$ = 26) of assessments but not useful in 41% ($n$ = 31). In contrast, when users were asked to rate helpfulness on a 5-point scale from 'very helpful' to 'not helpful at all', 45% ($n$ = 34) rated the tool as 'very helpful' and a further 32% ($n$ = 24) rated it as 'quite helpful'.

Positive responses from staff included that the tool was good for highlighting areas of importance or difficulties faced by the service user and/or the caregiver, for directing the reasons for the assessment, and allowing a more in-depth review of an individual's outcomes. Reasons for negative responses included considering that it had not influenced their practice as their clinical requirements would have been provided anyway that the goals had already been discussed, or that the client was well-known to staff. In some cases, a specific practical aspect, such as the child having outgrown the chair or that a footplate needed adjustment, was not picked up despite 'comfort' being an option for selection on the PCOM. Some expressed concerns that the goal-setting might be biased if the client had difficulty doing this independently, or that the possibility that use could raise expectations.

Staff were asked to report whether using the tool had impacted their eventual choice of prescription. In around a quarter of the assessments made by staff, it was felt that the use of the tools had affected their eventual prescription choices in 23% (12 of 53) of adults and 27% (6 of 22) of the children assessed. Where an impact was noted, reasons included the choice of power pack, the type of seating, a specific power chair, height adjustable handles for carer needs and the weight of the chair.

*"At the moment client is using S/P [self-propelled] manual W/Ch [wheelchair] but client will be limited with his independence and will impact his goal to have work and be pain-free and might develop worst pain on wrist, back and shoulder if not given a powered chair" (Site A, adult)*

*"Added height adjustable handles due to mother's own health issues" (Site D, child)*

The main reasons stated for a lack of impact included that the tools had little impact on clinical decisions, led to the same outcomes as the usual process or that any changes required were better identified through other means. A limited choice of wheelchair was also mentioned, especially in complex cases or due to other factors.

*"It was not possible to meet the health and well-being plan due to the criteria linked to 'Active wheelchair [Brand name] for people who require a wheelchair for outdoor use only" (Site A, adult)*

*"There are a limited number of buggies available which meet the SU's [service user's] clinical needs therefore this had little impact on the clinical decision made." (Site B, child)*

However, there was also an acknowledgment that the tool supported the prescription decision, even if unchanged.

*"Helped to direct reason for the assessment although same outcome would have been achieved" (Site B, adult)*

Among the users and their carers, positive comments included that the tools made it easier to explain needs to staff, allowed more time to think (if provided in advance) and encouraged everyone involved to think about things not previously considered.

*"To realise what you actually wanted. Things on there that I didn't think of" (Site A, adult)*

*"Made it easier to explain as clinician had a rough idea before appointment of my needs without having to explain lots of information. Completing it at home made it easier to fill in with time." (Site A, adult)*

Finally, staff were asked to state if use of the tool had made any difference to the amount of time taken in the assessment to use the tool, both for themselves alone and including any other staff involved. On average, the staff member completing the form estimated it added an extra 11 minutes to use the tool, range 0–30 minutes, median and mode were 10 minutes. Where they estimated any additional time spent by other staff, this increased to an extra 16 minutes of staff time in total where others were included, range 0–50 minutes (median 15 minutes, mode 10 minutes).

When looking at the eventual choice of how the equipment was to be financed, in the majority of cases (68%, $n$ = 51), this was stated to be notional NHS provision for both adults and children. For six adults (4 from Site A and one each from Sites B and D), a 'notional plus contribution' option was selected, and one adult from Site B selected the third-party option. In the latter case, the assessor felt it was the PWB paperwork that had assisted the selection of goals. For those selecting notional plus contribution, in three cases (one each from A, B and D), the assessor stated that the prescription had not been affected by use of the tool any more than their standard practice. In one service (Site A), it was felt that expectations of what could be provided under the notional financing were raised but could not be met. In two cases from Site A, the tool had been useful and had impacted on the prescription.

## Tool data

Consent to review their tool data was given by 67 service users or their carers, including 19 children. This provided an opportunity to assess the relative frequency of use of each item and to check that all were relevant (see S6 and S7 Files).

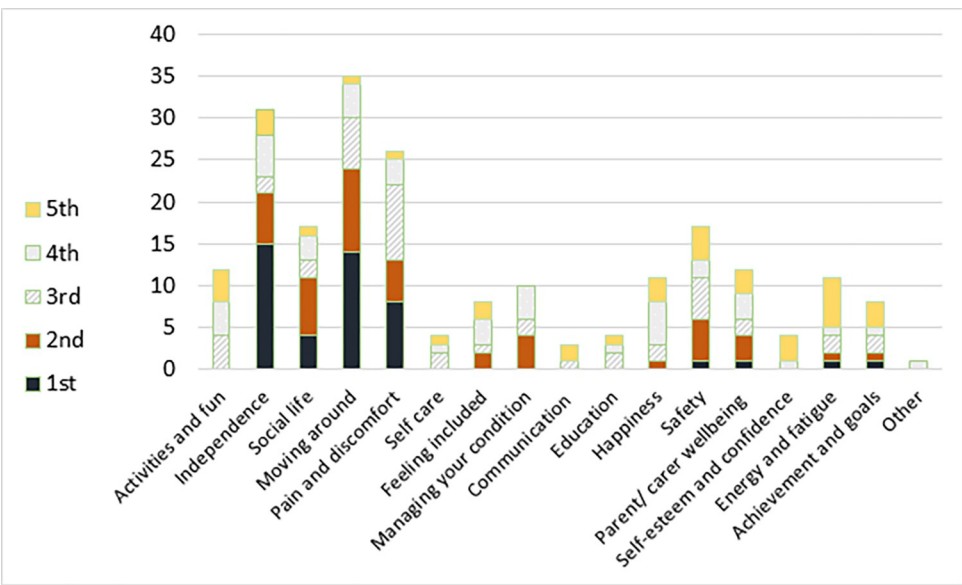

**Fig 1. Selection of WATCH-Ad outcomes by adults.**

Each of the items on the WATCh-Ad tool for adults were selected by at least one respondent in their top five, including the blank option 'other'. For the WATCh tool, all items except for 'Self-care', 'Happiness' and 'Achievement and goals' were selected at least once.

The most commonly selected items for adults were: 'Moving around', 'Independence', 'Pain & discomfort', 'Social life' and 'Safety'. For children, the most commonly selected were 'Moving around', 'Pain & discomfort', 'Social life', 'Activities & fun' and 'Education'. Figs 1 and 2 show the choices and relative rating by their order on the tool form (see Fig 1: Selection of WATCH-ad outcomes by adults and Fig 2: Selection of WATCH outcomes for children).

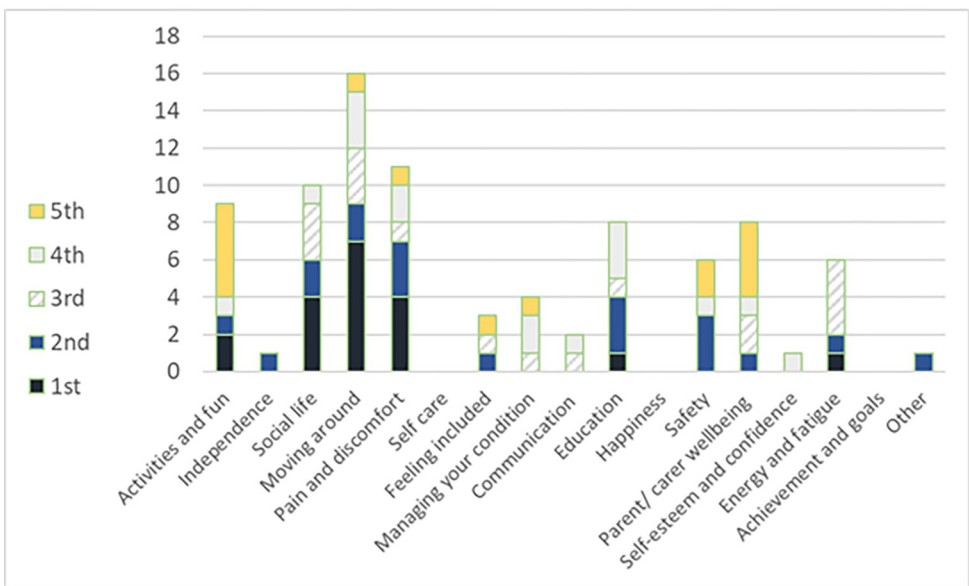

**Fig 2. Selection of WATCH outcomes for children.**

**2 - Independence** *n = 15*

- To be able move more and reach independently (2)
- To be more independent & be able to self propel short distance (2)
- want to take dog to park by himself…(2)

**4 - Moving around** *n = 14*

- I want to be able to get out on my own more often (2)
- Wheelchair to enable X to be taken outside (1)
- I want to be able to get from one room to another and to get out
  into the garden or to go out further (2)

**5 - Pain and Discomfort** *n = 8*

- Freely moving with less injuries (1)
- Don't want to be in pain as [x] sits in chair all day (2)

**3 - Social life** *n = 4*

- To sit in an upright position in order to interact c family and friends (2)
- Is important for me to spend time with people (2)

**1 - Activities and Fun** *n = 2*

- Be able to walk the dogs, complete jigsaws from chair (2)

**12 - Safety** *n = 1*

- To feel confident in the safety of the wheelchair (3)

**13 - Carer Wellbeing** *n=1*

- Old chair heavy and cumbersome (for carer and wife) (2)

**15 - Energy and fatigue** *n = 1*

- Be able to travel further without becoming tired and breathless (n/s)

**Fig 3. Reasons for choice of top outcome item chosen (satisfaction level): Adults.**

Figs 3 and 4 give examples of statements made by users or their carers against the most commonly selected outcomes chosen. Reviewing the description of the goals highlighted some overlap between areas, e.g. statements relating to 'moving around' might pick up aspects others might have felt related to 'safety' or 'activities'.

**4 – Moving around** *n = 7*

- Getting around safely (2)
- At home X bottom shuffles or walks a few steps holding onto furniture. Outside X uses his wheelchair for any distance further than a few meters [sic]. He uses a frame if a very short distance (3)

**5 - Pain and Discomfort** *n = 4*

- To be comfortable in his wheelchair (1)
- For distances when in pain. (1)

**3 - Social life** *n = 4*

- Feel happy to be around family and to spend time with them. Is important for me to spend time with people (5)
- Safety (sic) accessing different places (1)
- Can go out with family more (4)

**1 - Activities and Fun** *n = 2*

- Play with items while sat (no tray) (2)

**15 – Energy and Fatigue** *n = 1*

- To feel less tired and be able to rest (1)

**10 – Education** *n = 1*

- Access to/from school (n/s)

NB Items in top 10 but not 1st choices
13 - Parent/Carer wellbeing,
12 - Safety,
8 - Managing your condition
7 - Feeling included

**Fig 4. Reasons for choice of top outcome item (satisfaction level): Children and young people.**

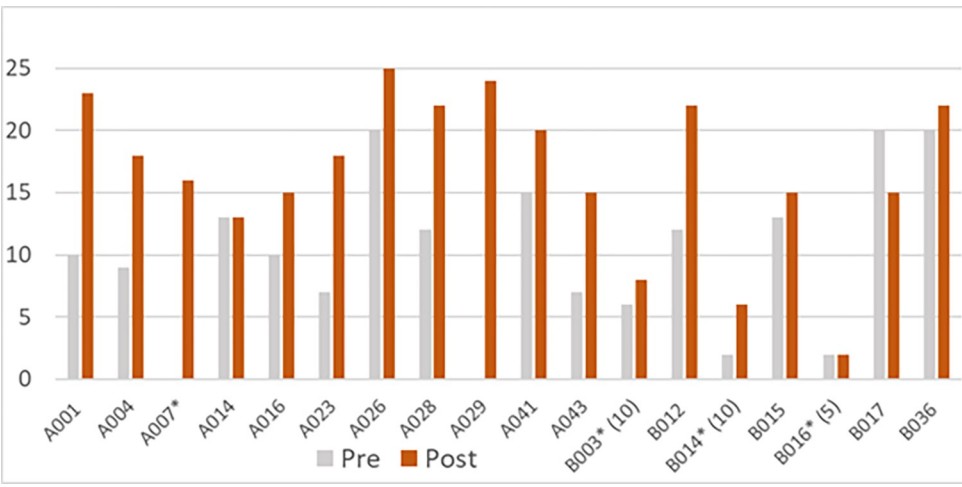

**Fig 5. Assessment and follow up scores at Sites A and B.**

## Follow-up data

The impact on study timings due to the COVID-19 pandemic meant that it was only possible to obtain follow-up data from a small number of respondents at Sites A and B. At site A, 15 of 26 (58%) participants had been provided with equipment, eleven of whom (73%) had provided follow up scores. At Site B, eight (53%) of fifteen participants available for follow-up provided follow up scores, of whom seven had provided formal consent. Assessment and follow-up scores for the total eighteen are presented in Fig 5 (see Fig 5 Assessment and follow up scores at Sites A and B). Where no follow up data was available, this was still being sought, or the user had not yet received their equipment. Reasons for the latter included not having decided on a PWB option, lack of availability of equipment or delays in attendance due to COVID-19 isolation. The satisfaction scores for patients able to be followed up during the study period were numerically higher than the assessment score for the majority of those for whom data were available. In only one case was the satisfaction score lower at follow up than at the assessment, and in one case the scoring was unchanged.

## Interviews

Interviews were semi-structured, aiming to investigate further the responses provided in the surveys. For service users, this was aimed at their own personal experience. For staff this aimed to cover all experiences of using the tool, including any use in their practice after the study period, with a focus on the practicalities. All interviews were carried out by telephone, by the lead author.

## User interviews

Interviews were carried out with two service users in January 2021. Both were aged in their 70s and from Site B/C. They stated that they had not had a previous wheelchair assessment before that in the study. These took place in March 2020, with equipment delivered in May 2020.

One interviewee was a woman who had been using a chair for three years following a stroke. On her survey she had responded positively toward the tool, stating it had taken 10 minutes to complete with assistance and that it was very helpful. The staff member completing their survey indicated it had added 15 minutes to the assessment. They had not rated usefulness but did comment that use had focused the assessment on the user's needs.

The user's key outcomes (satisfaction rating) were Safety (3), Pain and discomfort (3), Feeling included (2), Moving around (2), and Independence (2). Although she had not stated what she wanted for the last two on the tool, at the assessment and in the interview, she described feeling excluded and that people were looking down on her. Addressing the issues of wheelchair weight and an awkwardly positioned handbrake meant that she felt safer and far more positive about using her wheelchair:

*"It's a godsend, it really is, I'm not so frightened in it, as I was in the other one. "*

She felt the tool had been easy to use, relevant and didn't feel anything needed changing.

*"It was easy to understand. I could relate to what I used the wheelchair for. . .Especially the safety bit and how you feel in the chair. . ."*

The second interviewee was a man who had been using a wheelchair for two years post-amputation. He was also positive about the tool itself, stating on the survey that it had been easy to understand and complete, although it had been filled in for him by his wife. The staff member's survey indicated that use added 15 minutes to assessments but had assisted in setting goals and highlighted areas of importance to the client. The user's key outcomes (satisfaction rating) were Activities (2), Independence (2), Moving around (1), Happiness (2) and Self-esteem (2). His wife was present during the interview and was invited to contribute by the user. He commented:

*"When we filled everything in for me [sic] wheelchair, until you've had it, you don't really know what you do and don't want."*

## Staff interviews

Interviews with staff included four clinicians who had used the tools in assessments themselves, including after the end of the study period: the manager/clinician from Site A (*n* = 20), the clinical lead for Sites B/C (*n* = 14) and a clinician from Site B (*n* = 11) and a clinician from Site D (*n* = 16). The PWB liaison officer from Site A who was responsible for administering the WATCh or WATCh-Ad in advance prior to 35 assessments, and the clinical operational lead from Site C, who had managed staff responsible for 14 assessments using the tool, were also interviewed.

Overall, they could see the benefits of using the tools to provide a structured approach and allowing the patient voice to be heard:

*"Brilliant to take focus on and empower users–That is one of the real, real positives. . . gives them a voice. . .it's about empowering our service users so they're leading the assessment rather than being passive. . ." (Site B, clinician)*

The tools could assist discussion around PWBs although in some sites there was little experience of that:

*". . .feel it could be quite useful. The PWB process is to talk about options for them, so they don't have to go with our recommended provision. . . I can see there could be examples where it could be, for example if it picked up something that couldn't be achieved with NHS provision, but I haven't had that in practice to date." (Site D, clinician)*

Potential issues in use were around the additional workload on staff and time taken to include use of the tool, particularly where staff were dealing with PWB implementation, and in some cases COVID:

*". . . if the client didn't turn up on time or was late then clinicians were finding that they were constrained anyway in clinic time, so having to sit and try and complete this tool before the clinic appointment then made their assessment rushed" (Site A, PWB liaison)*

While assessments with children and younger people went particularly well, older users and particularly those living in care homes had more difficulties:

*"The people that seemed to show the biggest response was you know children and young people whose cognition was such that they can participate, they really valued I think being asked. . .. we had a thirteen-year-old in on Monday and he quite valued the process I would say" (Site B/C, clinical lead)*

*". . . residential homes, where there's frequent staff working with that individual and they're time limited with what time they can spend with that person. . . they weren't very, engaging in it. . ." (Site D, clinician)*

Staff all considered that time was needed to embed the process into their own ways of working, and in some cases, the pandemic had affected this:

*". . .you just have to get comfortable so that it's seen as a normal part of the assessment and not just an add-on. . .the WATCh documents have to be completed and then you find your own mojo as to how that's incorporated into the assessment." (Site B/C, clinical lead)*

Suggestions for maximising efficiency in use of the tool were also sought. By the end of the study, sites reported providing the tools in advance where possible, rather than to the service user 'blind' in clinic. This gave service users and their carers time to consider which outcomes were most important to them and discuss them with others. It also reduced the time spent in clinic.

Some staff raised concerns that tools might raise unrealistic expectations about the equipment likely to be provided or the benefits likely to be achieved. They noted that in a small number of cases, users did not engage with the tool as they '*just wanted a wheelchai*r'. Others considered some of the outcomes listed to be insensitive, for example, to a service user with a progressive or terminal illness. Some suggested that, where some patients struggled to select five options, three might be more appropriate. Some staff felt that the tools were more appropriate for new referrals than for users who were already familiar to the service.

## Discussion

This first study to assess the use of a PCOM across several providers alongside the introduction of PWBs demonstrated the ability of the tools to help users and clinicians to identify key outcome areas. The practicalities of use and the resource implications of introducing a PCOM into the PWB process were also assessed.

Kenny and Gowran's critical review identified several of the outcomes measures noted above and summarised their administrative burden, including the claimed time taken to complete [9]. This ranged from up to 15 minutes for the QUEST and the FEW, 30 mins for WhOM and 1 hour for the GAS. The PIADs is reported to take only 5–10 minutes, although

they cite a paper where use took 25 minutes. Thus, the average time reported by staff and users to complete the WATCh and WATCh-Ad tools is in line with the tools that are quickest to complete. However, it is acknowledged that the WhOM also incorporates some clinician-specific questions related to equipment provision, whereas the WATCh and WATCh-Ad can be used with any clinical assessment used by the service provider. Work describing the face and content validity of the WSQ did not calculate the average time taken to complete as it was felt that a mean would not represent a skewed distribution [18].

The real-world study presented here gave an insight into how organisations adapted their use of the tools, alongside gaining familiarity with their use. Based on this, it is recommended that, where possible, the WATCh and WATCh-Ad and information on the PWB process should be provided in advance with the invitation to the assessment appointment, whether paper-based or electronic.

The majority of responses from service users indicated that they had no problems or questions with the tool. Assessors noted problems or questions in almost half of their survey responses, but the majority were resolved during the assessment. Particular issues arose with users who had problems with the written format, for example, those without English as a first language or who had issues with mental capacity, sight or literacy. Similarly, those with hearing difficulties had issues if the form was being read over the phone.

With regard to suggestions to reduce the number outcomes to be selected by users, it should be noted that even in this small-scale study, all options listed on the WATCh-Ad were selected by or on behalf of at least one service user, including the open 'Other' option. Similarly, in the pilot work undertaken as part of the development of the original WATCh for children's tool, all items were selected by at least one child or their parent/guardian [7]. Whilst some of the reasons given in Part B could relate to a different area of Part A than was actually selected, this was not consistent enough to suggest any particular outcome listing was superfluous. If anything, the reasons given around the most commonly selected 'moving around' outcome often related to one of the other areas and could be considered redundant. The tools are intended to assist service providers in identifying the outcomes of most importance to a user by prompting the user to think about wider aspects of their need for a wheelchair and allow their reasoning to be recorded to refer back to. Several of the users commented spontaneously that the list made them think of other aspects of their need for a wheelchair and enabled them to discuss these with the staff. As patient-centred rather than patient-reported outcome measures, the WATCh and WATCh-Ad obtain scores to be used for intra-individual comparisons of satisfaction before and after receipt of the equipment and are not intended to be used as a comparison between users, services or equipment. Where staff consider it appropriate or necessary in individual cases, selecting three outcomes might be sufficient to allow the assessment to focus on those most important to the user. The authors consider that wording the instructions to state that they can choose three to five options from the list might allow staff some leeway to discuss the user's needs in more depth and focus. It would still be possible to use the scoring tool to assess a percentage change in scores.

The type of assessment situation in which the tools should be implemented may need refining. Staff found them unnecessary in situations where straight-forward repairs or adjustments only were required. Early screening and triage of referrals and contact with users should be able to identify cases where the tool would not be necessary, and some individual service users or carers commented that staff already knew more about their needs and circumstances than tools could identify. However, as 77% of the users covered by the assessments stated that the tools were helpful or very helpful, this indicates that the tools helped them feel more involved in the decision-making even if the service's choice of equipment was unchanged. We suggest that the tools should be offered to existing as well as new users but that the accompanying

information for users be worded to indicate that it is a standard approach being taken by the service, even for those users who were already well-known to them.

In this study, the majority of service users providing feedback were positive about use of the WATCh tools in helping them consider their wider needs and discuss these with their wheelchair service provider. In addition, use of the tools had influenced the prescription and the PWB choice in up to 33% of cases at some sites. Where staff reported that the tool had not been completed by users, the reasons were varied and mostly practical and could be addressed by the provision of the tool and appropriate accompanying information in advance. A statement as to why a general tool is being used by the service may help explain its use with service users already well known to the service, and also reduce issues of insensitivity. A reminder to bring to clinic could be added to any reminder communications. Further developments of the tools should include translations tested among key populations, availability in a digital format, and ability for information from use of the tools to be automatically linked to the service user's records.

The uptake of any new treatment or process is reliant on successful implementation. Proctor et al. (2009), in work relating to mental health services in the United States, referred to three kinds of outcomes in implementation research–client outcomes, service outcomes and implementation outcomes. Client outcomes were defined as satisfaction, function and symptomatology, with service outcomes based on the six Institute of Medicine (IoM) standards of Care published in 2001, namely efficiency, safety, effectiveness, equity, patient-centeredness and timeliness. Implementation outcomes were further developed into eight conceptually distinct outcomes of acceptability, adoption, appropriateness, feasibility, fidelity, implementation cost, penetration, and sustainability [29].

The WATCh and WATCh-Ad PCOMs are designed to uncover areas of importance to individual patients around satisfaction and function, but only to a certain extent symptomatology. This study looked at service outcomes plus the implementation outcomes of acceptability, appropriateness, and feasibility in order to make recommendations to maximising successful adoption, fidelity, uptake and sustainability. Further work is required to fully assess the relationship between costs and benefits of implementing a PCOMs / PWB process on wheelchair provision.

The study also allowed assessment of the aspects of validity relevant to PCOMs [30], which are used for intra-person evaluation over time and not intended for use in a comparative test: content validity, face validity and clinical validity.

Content validity relates to the ability of a tool to measure what it is supposed to measure. The initial WATCh tool was developed through previous academic work with children and young people, and, based on work on quality of life in people with mobility impairments by the authors, was deemed relevant to be adapted for use with adults. Although some participants in this study felt the number of items was too long, all outcomes listed were selected as important by at least one participant.

Face validity refers to the acceptability of the test to the test taker. The present study added information on adult service users and their carers to the information gained from the work developing the WATCh for children [7] and found that use of the tool was considered helpful or very helpful to a majority of participants. However, further work is needed to increase usability among service users with additional needs. Ease of use could be increased further by enhancing the ability for the tools to be shared between the service user and the service by electronic means.

Clinical validity is whether a measure has overall usefulness in a clinical situation. In this study, there was a slight majority of assessments where staff did not feel that the tool had been

helpful or had altered the prescription. However, an impact was noted in 24%, suggesting that one in four patients may receive benefit over and above the usual assessment process.

The authors acknowledge the potential limitations of the work. This relatively small study aimed to review the use of the tools in varied situations within a short timescale. From a practical perspective, more time was needed to formally pilot the survey tools. However, the study team, including clinicians, commissioners and a service user reviewed the protocol and documents. In hindsight, it would have been preferable to include more options around the level of usefulness to staff in place of a yes/no option. **The authors acknowledge that, given the timescales, it was not possible to perform a test-retest of satisfaction scores, both pre- and post provision of the equipment.**

As described above, the impact of the COVID-19 pandemic on the NHS in terms of capacity and ability to perform research affected the timing, and thus, the number of assessments able to be reported upon, particularly the ability to obtain more in-depth views from the service users. Although many assessments at sites A to C took place before the announcement to stop providing all but emergency provision in the NHS, there had already been significant media interest in the pandemic, so those attending clinic may have been the most severe or urgent cases. Even after research could be restarted, significant concerns about the risk of infection remained. This may have compounded the differences between Site D and the other sites in addition to the type of service provision and experience of use of PWB or the WATCh and WATCh-Ad, as there was some reluctance to attend clinic or have visitors to the home.

Changes in the numbers of wheelchair service users in the quarterly data provided by CCGs in England suggest an impact of the pandemic on service provision. In October-December 2019, the last quarterly data obtained before collection was paused, over 700,000 users of wheelchair services were registered, including over 75,000 children [31]. In March 2022, the figures were over 600,000 and 62,000, respectively [32], and the latest available data show that numbers have still not reached pre-pandemic levels.

Measures taken to reduce transmission of the virus also led to the implementation of different ways of working, for example, making more use of telephone screening and/or digital technology for screening/triage and digital technology for remote assessment and use of home visits. Communication via smartphone or email was introduced to speed up the time for appointment notifications to be received by service users. However, there is evidence that some of these new ways of working may be in use for longer than just the immediate period of the pandemic; hence, information gained on benefits and issues relating to use of the WATCh and WATCh-Ad alongside these in this study will remain relevant going forwards.

It was hoped that the use of such tools would increase the opportunity for the voices of the service users to be heard. Due to the circumstances of many service users, responses were obtained through the carer and, in some cases, with the help of the assessor. The original WATCh was developed for children to complete themselves. In this study, while the results from the staff interviews noted that the tools were generally well received by children and young adults, all the survey respondents completing the WATCh for children were parents or carers. In many cases there will have been practical, physical or capability reasons why a child could not respond for themselves. However, it will be important that assessors ensure that children and young people, and indeed adults where a carer has provided the responses, agree with the outcomes selected on their behalf as far as possible.

The addition of any new practice into an existing way of working would be expected to increase the time involved, at least until a service assesses the most efficient ways of incorporating it into standard practice. This can be achieved through staff training and mentoring and by encouraging staff to develop their own way of working with it within the protocol. Practical suggestions noted above include providing the tool in advance with appropriate instruction

and developing ways of using the tools alongside new digital technologies which are already increasing partly due to the pandemic.

## Conclusions

The majority of participating wheelchair service users stated that the WATCh and WATCh-Ad PCOMs were helpful, assisting them in identifying their required outcomes and discussing these with staff during their assessments. Staff noted that use of the tools impacted prescriptions or assisted in the choice of PWB in just under a quarter of their assessments, a finding which, if rolled out nationally, would extrapolate to many service users. As with any change, service providers will need time to incorporate the tools into their standard practice. Provision of the tools to users in advance of appointments allow users time to consider their needs and reduce the time in clinic. Further work is needed to broaden the accessibility and service utility of the tools, for example, translations, digital availability and automatic linking of tool scorings into service users' records in existing clinical systems.

## Supporting information

**S1 File. WATCh-Ad assessment and follow-up forms.**
(PDF)

**S2 File. WATCh assessment and follow-up forms.**
(PDF)

**S3 File. Data collection materials.**
(PDF)

**S4 File. Flowchart of participants' data collection.**
(PDF)

**S5 File. Additional figures.**
(PDF)

**S6 File. Survey data.**
(XLSX)

**S7 File. WATCh and WATCH-Ad tool data.**
(XLSX)

## Acknowledgments

The authors wish to thank the study team, the participating staff and service users at the sites involved, the CCGs who supported the study, and Dr Catherine Lawrence, Bangor University, for reviewing the original manuscript.

## Author Contributions

**Conceptualization:** Lorna Tuersley, Rhiannon Tudor Edwards, Nathan Bray.

**Data curation:** Lorna Tuersley.

**Formal analysis:** Lorna Tuersley.

**Funding acquisition:** Lorna Tuersley, Rhiannon Tudor Edwards.

**Investigation:** Lorna Tuersley.

**Methodology:** Lorna Tuersley, Nathan Bray.

**Project administration:** Lorna Tuersley.

**Supervision:** Rhiannon Tudor Edwards, Nathan Bray.

**Visualization:** Lorna Tuersley, Naa Amua Quaye.

**Writing – original draft:** Lorna Tuersley.

**Writing – review & editing:** Kalpa Pisavadia.

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
