## [Decision Letter · Decision Letter 0]

16 Jan 2024

PONE-D-23-37995Use of patient-centred outcome measures alongside the personal wheelchair budget process in NHS England: a mixed methods approach to exploring the staff and service user experience of using the WATCh and WATCh-AdPLOS ONE

Dear Dr. Pisavadia,

Thank you for submitting your manuscript to PLOS ONE. After careful consideration, we feel that it has merit but does not fully meet PLOS ONE’s publication criteria as it currently stands. Therefore, we invite you to submit a revised version of the manuscript that addresses the points raised during the review process.

We look forward to receiving your revised manuscript.

Kind regards,

Yih-Kuen Jan, PhD

Academic Editor

PLOS ONE

Journal Requirements:

"I have read the journal's policy and the authors of this manuscript have the following competing interests: the funding for the the lead author for the study was received from NHS England who had commissioned the initial development of the PCOM study evaluated in the study."

5. We noted in your submission details that a portion of your manuscript may have been presented or published elsewhere. [The previous submission to PLOS was pre-published We would wish this to be withdrawn.] Please clarify whether this [conference proceeding or publication] was peer-reviewed and formally published. If this work was previously peer-reviewed and published, in the cover letter please provide the reason that this work does not constitute dual publication and should be included in the current manuscript.

6. In this instance it seems there may be acceptable restrictions in place that prevent the public sharing of your minimal data. However, in line with our goal of ensuring long-term data availability to all interested researchers, PLOS’ Data Policy states that authors cannot be the sole named individuals responsible for ensuring data access (http://journals.plos.org/plosone/s/data-availability#loc-acceptable-data-sharing-methods).

7. Please amend your list of authors on the manuscript to ensure that each author is linked to an affiliation. Authors’ affiliations should reflect the institution where the work was done (if authors moved subsequently, you can also list the new affiliation stating “current affiliation:….” as necessary).

8. Please include your full ethics statement in the ‘Methods’ section of your manuscript file. In your statement, please include the full name of the IRB or ethics committee who approved or waived your study, as well as whether or not you obtained informed written or verbal consent. If consent was waived for your study, please include this information in your statement as well. 

9. Please upload a copy of Supporting Information Figure/Table/etc. S5 Dataset which you refer to in your text on page 35.

Reviewers' comments:

Reviewer's Responses to Questions

**Comments to the Author**

1. Is the manuscript technically sound, and do the data support the conclusions?

Reviewer #1: Yes

Reviewer #2: Partly

2. Has the statistical analysis been performed appropriately and rigorously? 

Reviewer #1: N/A

Reviewer #2: No

3. Have the authors made all data underlying the findings in their manuscript fully available?

Reviewer #1: Yes

Reviewer #2: Yes

4. Is the manuscript presented in an intelligible fashion and written in standard English?

Reviewer #1: Yes

Reviewer #2: Yes

5. Review Comments to the Author

Reviewer #1: This study aimed to assess the use of the WATCh and the WATCh-Ad assess for outcome achievement in wheelchair user and service user. This paper is the useful study that can improve the assessment and the PWB service. I think this manuscript should be accepted. However, there are little suggestion to improve the rationale and gap of this study. Specific comment for each section is below:

Method

Page 15 Line 158 : Is the author have the evident about the validity and reliability of WATCh and the WATCh-Ad ? Please present the psychometric properties of the questionnaire. and it can be used as the evidence in discussion part.

Page 19 Line 262 : For quantitative analysis, why this study was not use comparative statistics for analysis score between pretest and follow up period?

Results

Page 24 Line 336-338: Please add percentage after number in the problems that were reported

Discussion

Page 36 Line 617 : In the result section, there are report some of the disadvantage and limitation of WATCh and the WATCh-Ad. Please discuss more information to solve this problem and how to apply in service user.

Reviewer #2: Part introduction :

- To ensure that your outcomes aligns seamlessly with your aim. Should add any specific details regarding the desired outcomes and how they will be measured.

Part method :

- The incorporation of a flowchart is recommended to facilitate a clearer understanding of the study protocol among readers.

- Is there a potential impact on the study results by eliminating certain groups of samples? It's important to note that samples from each location represent distinct populations, and their exclusion may have implications for the overall findings.

- Comprehensive information about data analysis, with a focus on addressing rigor and trustworthiness issues, should be included.

6. PLOS authors have the option to publish the peer review history of their article (what does this mean?). If published, this will include your full peer review and any attached files.

Reviewer #1: No

Reviewer #2: No

---

## [Author Response · Author response to Decision Letter 0]

11 Mar 2024

Response to Editor’s comments

This has been addressed.

2. Consider depositing your raw data in a repository.

Participants could be identifiable from the level of detail required within the information due to small number of sites and subjects.

3.1. ‘Funding Information’ and ‘Financial Disclosure’ sections do not match.

This has been addressed.

3.2. Provide the correct grant numbers for the awards you received for your study in the ‘Funding Information’ section.

Not applicable - no grant numbers ot Addressed.

Provided in cover letter.

5. Please clarify whether this [conference proceeding or publication] was peer-reviewed and formally published.

Addressed in cover letter

6. Please also provide non-author contact information (phone/email/hyperlink) for a data access committee, ethics committee, or other institutional body to which data requests may be sent.

As Item 2

7. Amend your list of authors on the manuscript to ensure that each author is linked to an affiliation.

This has been addressed.

8. Please include your full ethics statement in the ‘Methods’ section of your manuscript file. In your statement, please include the full name of the IRB or ethics committee who approved or waived your study, as well as whether or not you obtained informed written or verbal consent.

This has been addressed.

9. Please upload a copy of Supporting Information Figure/Table/etc. S5 Dataset which you refer to in your text on page 35.

Provided as S6 and S7 (New S4 flowchart)

10. Please review your reference list to ensure that it is complete and correct. If you have cited papers that have been retracted, please include the rationale for doing so in the manuscript text, or remove these references and replace them with relevant current references.

This has been addressed.

Response to Reviewer 1’s comments

1. Method Page 15 Line 158 : Is the author have the evident about the validity and reliability of WATCh and the WATCh-Ad ? Please present the psychometric properties of the questionnaire. and it can be used as the evidence in discussion part.

The WATCh and WATCh-Ad are intended as tools for use by clinicians in their assessment of an individual wheelchair user's needs, along with the user or their carer, based on aspects of their life that could be impacted by their personal circumstances and need for equipment. They are not intended to be used to make comparisons between groups of users, different services or equipment, for which other measures are available; we have attempted to make this clear in the paper (see lines 604-607). The outcomes areas were determined from previous work (see references #7 and #8).

2. Method Page 19 Line 262 : For quantitative analysis, why this study was not use comparative statistics for analysis score between pretest and follow up period?

The scoring system aims to indicate any change in the level of satisfaction before and after receipt of the equipment for an individual service user. They are not appropriate for use to make comparisons between groups of users, different services or equipment, for which other measures are available; we have attempted to make this clear in the paper (see lines 604-607). For this reason and because of the small number of users receiving their equipment in the time period of the study it was not felt appropriate to carry out statistical analysis, but to present the scoring data. Further comparative analysis is beyond the scope of this study and would not be in-keeping with the purpose of the tools.

3.1. Results Page 24 Line 336-338: Please add percentage after number in the problems that were reported.

This has been addressed.

3.2. Discussion Page 36 Line 617 : In the result section, there are report some of the disadvantage and limitation of WATCh and the WATCh-Ad. Please discuss more information to solve this problem and how to apply in service user.

Lines added into discussion (new lines 639 – 646) and conclusion (line 736)

Response to Reviewer 2 comments

1. Part introduction : To ensure that your outcomes aligns seamlessly with your aim. Should add any specific details regarding the desired outcomes and how they will be measured.

This study was not aiming to obtain desired outcomes but to gain insight into the usability. Last paragraph of introduction has been revised to reflect this (lines 45 – 46).

2. Part Method - The incorporation of a flowchart is recommended to facilitate a clearer understanding of the study protocol among readers.

Flowchart added as new Supplementary File 4

3. Is there a potential impact on the study results by eliminating certain groups of samples? It's important to note that samples from each location represent distinct populations, and their exclusion may have implications for the overall findings.

No groups of samples were omitted - the differences in numbers of staff surveys, user surveys and user tool data analysed was where data was not available or participants did not consent to use of particular data. We therefore believe that this comment is beyond the remit of the study.

4. Comprehensive information about data analysis, with a focus on addressing rigor and trustworthiness issues, should be included.

As noted in 1) the study aimed to obtain descriptive statistics on acceptability of use of the tool use among users and staff to obtain information on users' key outcomes and score before and after receipt of their equipment. We therefore feel that the request to report comprehensive information about data analysis, focussing on rigor and trustworthiness, is beyond the stated remit of this study.

---

## [Decision Letter · Decision Letter 1]

22 Apr 2024

PONE-D-23-37995R1Use of patient-centred outcome measures alongside the personal wheelchair budget process in NHS England: a mixed methods approach to exploring the staff and service user experience of using the WATCh and WATCh-AdPLOS ONE

Dear Dr. Pisavadia,

Thank you for submitting your manuscript to PLOS ONE. After careful consideration, we feel that it has merit but does not fully meet PLOS ONE’s publication criteria as it currently stands. Therefore, we invite you to submit a revised version of the manuscript that addresses the points raised during the review process.

We look forward to receiving your revised manuscript.

Kind regards,

Yih-Kuen Jan, PhD

Academic Editor

PLOS ONE

Journal Requirements:

Reviewers' comments:

Reviewer's Responses to Questions

**Comments to the Author**

1. If the authors have adequately addressed your comments raised in a previous round of review and you feel that this manuscript is now acceptable for publication, you may indicate that here to bypass the “Comments to the Author” section, enter your conflict of interest statement in the “Confidential to Editor” section, and submit your "Accept" recommendation.

Reviewer #1: All comments have been addressed

Reviewer #2: All comments have been addressed

Reviewer #3: (No Response)

2. Is the manuscript technically sound, and do the data support the conclusions?

Reviewer #1: Yes

Reviewer #2: No

Reviewer #3: Partly

3. Has the statistical analysis been performed appropriately and rigorously? 

Reviewer #1: Yes

Reviewer #2: No

Reviewer #3: No

4. Have the authors made all data underlying the findings in their manuscript fully available?

Reviewer #1: Yes

Reviewer #2: Yes

Reviewer #3: No

5. Is the manuscript presented in an intelligible fashion and written in standard English?

Reviewer #1: Yes

Reviewer #2: No

Reviewer #3: No

6. Review Comments to the Author

**Reviewer #1**: The authors have addressed the comments with reasonable reason and evidence. In addition, This paper has the useful information and scientific writing that can improve the assessment of outcome achievement in wheelchair user and service user. In my opinion, this manuscript should be accepted.

**Reviewer #2:** It appears that the outcomes of the study have not sufficiently addressed the defined objective, specifically in generating guidance for use alongside the PWB process.

**Reviewer #3:** It is great to see an outcome measurement tool included in service provision to improve the provision quality. The manuscript has strengths and weaknesses. The strengths are that the tool demonstrates feasibility of use and implementation of the tool in clinical settings. This is demonstrated through staff and user testing in this study. The weaknesses lie in the organization and conciseness of the manuscript, lack of methodology to study the aim, and poor literature review. Review comments:

Introduction: The manuscript lists several mobility assessment tools however, there is little comparison on how these tools measure satisfaction related to mobility. For children, the FMA-FC tool was developed in early 2018. The MOBQOL is developed for QOL assessment. Please include these tools in the review and synthesize why the WATCH is piloted in this study than other validated tools.

Psychometric properties of the tool: Please verify if this tool has undergone test-retest reliability testing. If not, please note it as a limitation. This means the responses to the tool outcome items will vary across times for a user and across users with same outcome levels.

Main Aim: As stated on line 134, the main study aim is to compare satisfaction ratings before and after. The question is HOW? Please provide methodology and statistical testing for the same.

Table 1 should be in results. The COVID-19 related content does not seem relevant.

Line 266: Please describe the method to transform qualitative data into themes.

The way results are arranged under subheadings, could you arrange the methods and analyses.

Line 330: separate the mean times for the two cohorts.

Discussion: Please begin this section with the highlights of the study - usability and acceptance of the tool. Was your main aim addressed? The time assessment noted in first parah can follow.

The validity text from 649-670 is redundant as far as the main aim is concerned.

I hope these comments help in improving the manuscript.

7. PLOS authors have the option to publish the peer review history of their article (what does this mean?). If published, this will include your full peer review and any attached files.

Reviewer #1: No

Reviewer #2: No

Reviewer #3: **Yes: **Anand Mhatre

---

## [Author Response · Author response to Decision Letter 1]

5 Jul 2024

Response to Editor’s comments

Comment (abridged)

If you have cited papers that have been retracted, please include the rationale for doing so in the manuscript text, or remove these references and replace them with relevant current references.

Response/ Rebuttal

Assuming this relates to comments from Reviewer 3, the additional work they refer to is already is already cited – see rebuttal #1

Response to Reviewer 3’s comments

Comment (abridged) 1:

Introduction: The manuscript lists several mobility assessment tools however, there is little comparison on how these tools measure satisfaction related to mobility. For children, the FMA-FC tool was developed in early 2018. The MOBQOL is developed for QOL assessment. Please include these tools in the review and synthesize why the WATCH is piloted in this study than other validated tools.

Addressed/Rebuttal

The MoBQoL and FMA-FC are both already cited as references 8 and 16 respectively, however the authors thank the reviewer for highlighting that the FMA has been adapted for use with children as the FMA-FC, and that further explanation is required to explain why the WATCh tools offer a different approach to all those listed. New statements added at lines 95-96, 115, 138-142.

Comment abridged 2

Psychometric properties of the tool: Please verify if this tool has undergone test-retest reliability testing. If not, please note it as a limitation. This means the responses to the tool outcome items will vary across times for a user and across users with same outcome levels.

Addressed/Rebuttal

One of the aims of the WATCh and WATCh-Ad tools is to obtain a baseline measure of satisfaction with the aspects of life highlighted as of most importance to the wheelchair user. This is then compared with a re-score of satisfaction after use of the equipment provided subsequently. Therefore it would be anticipated that the scoring should change with time, hopefully improved, if the equipment has helped meet the user's needs.

Comment (abridged) 3:

Main Aim: As stated on line 134, the main study aim is to compare satisfaction ratings before and after. The question is HOW? Please provide methodology and statistical testing for the same.

Addressed/Rebuttal

The authors thank the reviewer for the comment and have explained that the scoring for the WATCh and WATCh-Ad tools is within-patient only, i.e. the clinician is only looking at any numerical change between assessment and post-equipment scores for that patient. A new statement is added at line 91-93 and summary findings presented at lines 477-480. As the tools are not intended to provide scoring between patients or to be used to compare performance of different equipment or different service providers etc., as each patient's situations and needs are unique.

---

## [Decision Letter · Decision Letter 2]

12 Aug 2024

PONE-D-23-37995R2Use of patient-centred outcome measures alongside the personal wheelchair budget process in NHS England: a mixed methods approach to exploring the staff and service user experience of using the WATCh and WATCh-AdPLOS ONE

Dear Dr. Pisavadia,

Thank you for submitting your manuscript to PLOS ONE. After careful consideration, we feel that it has merit but does not fully meet PLOS ONE’s publication criteria as it currently stands. Therefore, we invite you to submit a revised version of the manuscript that addresses the points raised during the review process.

We look forward to receiving your revised manuscript.

Kind regards,

Yih-Kuen Jan, PhD

Academic Editor

PLOS ONE

Journal Requirements:

Reviewers' comments:

Reviewer's Responses to Questions

**Comments to the Author**

1. If the authors have adequately addressed your comments raised in a previous round of review and you feel that this manuscript is now acceptable for publication, you may indicate that here to bypass the “Comments to the Author” section, enter your conflict of interest statement in the “Confidential to Editor” section, and submit your "Accept" recommendation.

Reviewer #1: All comments have been addressed

Reviewer #3: (No Response)

2. Is the manuscript technically sound, and do the data support the conclusions?

Reviewer #1: Yes

Reviewer #3: No

3. Has the statistical analysis been performed appropriately and rigorously? 

Reviewer #1: N/A

Reviewer #3: No

4. Have the authors made all data underlying the findings in their manuscript fully available?

Reviewer #1: Yes

Reviewer #3: Yes

5. Is the manuscript presented in an intelligible fashion and written in standard English?

Reviewer #1: Yes

Reviewer #3: No

6. Review Comments to the Author

Reviewer #1: This research can be published in this journal. This study can improve wheelchaired service for both of the staff and service user. The authors rewrite manuscripts and give more detail in results and the application of research that reviewer comments.

Reviewer #3: Thank you for addressing the comments. The second comment is about test-retest reliability which is a limitation of the tool before you perform usability testing in this study. I understand the response but that does not verify test-retest unfortunately. The third comment is about the lack of statistical analysis to test your aim which is not in your abstract methods and is present in the manuscript. See Aims: Our aim was to assess outcome achievement at the level of the patient, referred to here as the service user, by comparison of satisfaction ratings with their key outcome areas before and after provision of equipment, and identify aspects of use of the tools important to them and the service provider in order to maximise wider implementation. So, you are comparing satisfaction. In your Analysis section there is not statistical test for that. In the results line 466 onwards and Fig. 3, you simply say higher or lower scores but you want to test for statistical significance as that is your main aim.

7. PLOS authors have the option to publish the peer review history of their article (what does this mean?). If published, this will include your full peer review and any attached files.

Reviewer #1: No

Reviewer #3: No

---

## [Author Response · Author response to Decision Letter 2]

20 Sep 2024

Response to editors comment

Item 1

If you have cited papers that have been retracted, please include the rationale for doing so in the manuscript text, or remove these references and replace them with relevant current references.

Response 

N/A

Response to reviewer 3

Item 6

The second comment is about test-retest reliability which is a limitation of the tool before you perform usability testing in this study. I understand the response but that does not verify test-retest unfortunately. The third comment is about the lack of statistical analysis to test your aim which is not in your abstract methods and is present in the manuscript. See Aims: Our aim was to assess outcome achievement at the level of the patient, referred to here as the service user, by comparison of satisfaction ratings with their key outcome areas before and after provision of equipment, and identify aspects of use of the tools important to them and the service provider in order to maximise wider implementation. So, you are comparing satisfaction. In your Analysis section there is not statistical test for that. In the results line 466 onwards and Fig. 3, you simply say higher or lower scores but you want to test for statistical significance as that is your main aim.

Response

Thank you for clarifying your comments. 

Due to the way the tool is implemented with the user through discussion with the service provider, it was considered impractical to carry out a test-retest for the pre- and/or post- provision of equipment phases. Your comments have been noted and we have now added this limitation within the Discussion. 

Due to the nature of the study, the protocol planned for descriptive statistics. We do not state within the manuscript that a formal statistical analysis was performed. In response to your comment, we have further clarified in the results section of the abstract that results are simply presented graphically.

---

## [Editor Report · Decision Letter 3]

16 Oct 2024

Use of patient-centred outcome measures alongside the personal wheelchair budget process in NHS England: a mixed methods approach to exploring the staff and service user experience of using the WATCh and WATCh-Ad

PONE-D-23-37995R3

Dear Dr. Pisavadia,

We’re pleased to inform you that your manuscript has been judged scientifically suitable for publication and will be formally accepted for publication once it meets all outstanding technical requirements.

Kind regards,

Yih-Kuen Jan, PhD

Academic Editor

PLOS ONE
---

## [Editor Report · Acceptance letter]

26 Dec 2024

PONE-D-23-37995R3 

PLOS ONE

Dear Dr. Pisavadia, 

I'm pleased to inform you that your manuscript has been deemed suitable for publication in PLOS ONE. Congratulations! Your manuscript is now being handed over to our production team.

Kind regards, 

on behalf of

Dr. Yih-Kuen Jan 

Academic Editor

PLOS ONE